# Daytime Variation of Chloral Hydrate-Associated Sedation Outcomes: A Propensity-Matched Cohort Study

**DOI:** 10.3390/jcm12031245

**Published:** 2023-02-03

**Authors:** Yu Cui, Langtao Guo, Li Xu, Qixia Mu, Qunying Wu, Lu Kang, Qin Chen, Yani He, Hong Liu

**Affiliations:** Department of Anesthesiology, Chengdu Women’s & Children’s Central Hospital, School of Medicine, University of Electronic Science and Technology of China, Chengdu 610091, China

**Keywords:** daytime variation, chloral hydrate, sedation failure rate, adverse events

## Abstract

Background: Physiological processes influencing a drugs’ efficacy change substantially over the course of the day. However, it is unclear whether there is an association between the sedative success rate of chloral hydrate and the time of day. We conducted a retrospective study of 41,831 cases, to determine if there was a difference in sedation success rate with chloral hydrate in children seen in the morning and afternoon. Methods: Patients who accepted the sedation service were included. Eligible patients were divided into two cohorts of morning and afternoon cases, according to the time of day when the initial dose of chloral hydrate was administered. To ensure that the two groups were comparable, a propensity score matching method was utilized. Results: The success rate with the initial dose of chloral hydrate was higher in patients who received sedation services in the afternoon. In the subgroup analysis, the afternoon cases had a higher sedation success rate compared to the morning cases in male patients; whereas, in female patients, no difference was detected between the morning versus afternoon cases. Conclusions: These results show that the afternoon cases had a higher sedation success rate than the morning cases, despite the afternoon cases receiving relatively lower initial dose than the morning cases. However, the clinical significance remains to be discussed, and further prospective studies are needed to validate the findings.

## 1. Introduction

When infants and children undergo computed tomography (CT), magnetic resonance imaging (MRI), and electrocardiography, immobilization is necessary, to obtain high-quality images. It is best if the patient can sleep naturally, but this is not an easy task. Thus, most institutions prescribe sedatives to perform these procedures when pediatric patients are unable to cooperate.

Chloral hydrate is a nonopiate, nonbenzodiazepine sedative-hypnotic drug, which is absorbed from the gastrointestinal tract and rapidly converted to the pharmacologically active metabolite trichloroethanol (TCE), which is responsible for the sedative effects [1]. Although chloral hydrate is recommended as the first-line drug by the UK National Institute for Clinical Excellence (NICE) for procedural sedation of children under 15 kg [2], the use of chloral hydrate has declined significantly globally since these recommendations were issued. In many countries, chloral hydrate is no longer available unless specially compounded. However, it is also used in developing countries such as China [3] and Turkey [4], as well as in some developed countries (South Korea [5], Japan [6], Spain [7], and Canada [8]). However, concerns over the use of chloral hydrate have increased due to serious adverse events [9], while other authors still support the use of chloral hydrate for sedation outside the operating room [7,10], especially for non-anesthesiologist-led sedation. Furthermore, sedation failure rates associated with chloral hydrate have been reported to be up to 28% [11]. Risk factors associated with chloral hydrate sedation failure are heavier weight, patients with a history of sedation or sedation failure, and patients undergoing MRI or two or more procedures simultaneously, which have been well studied [3].

Physiological processes influencing a drugs’ efficacy change substantially over the course of the day, and various authors have summarized time-of-day variation in the action of anesthetic drugs in adults and pediatric patients [12]. Szolnoki et al. retrospectively analyzed 2340 children who underwent brain MRI under general anesthesia and found a strong relationship between time of day and the length of post-anesthetic care unit (PACU) stay. The length of recovery time after general anesthesia before 12:00 P.M. was shorter than that after 12:00 P.M. [13], indicating that circadian rhythms affected anesthetics efficacy. Theoretically, the sedative effect can be maximized when the child desires natural sleep, not only increasing the success rate of sedation but also prolonging the duration of sedation. However, it is unclear whether there is a relationship between chloral hydrate sedation success rate and time of day, although propofol-induced dysregulation of circadian rhythms was reported in an animal study [14]. Therefore, we hypothesized that administering chloral hydrate at the time of day when the efficacy of chloral hydrate was most effective would improve the success rate of sedation. To evaluate the effect of circadian rhythms on the success rate of chloral hydrate sedation, we performed a retrospective study of 41,831 cases, to determine if chloral hydrate-induced sedation success rates were different in children who were sedated in the morning vs. afternoon. Gender was considered in the subgroup analysis. Additionally, the safety of chloral hydrate sedation was also evaluated.

## 2. Materials and Methods

Children aged 29 days to 18 years who received sedation services for non-invasive procedures at Chengdu Women’s and Children’s Central Hospital between 1 December 2019 and 30 April 2022 were included. Ethical approval was obtained from the local institutional board of Chengdu Women’s and Children’s Central Hospital (No. 2022(56)). Written informed consent was waived due to the nature of the retrospective study.

### 2.1. Patient Selection and Sedation Methods

At our institution, the sedation center, managed by the anesthesiology department, is responsible for pediatric sedation during working hours, while the emergency department is responsible for sedation after working hours and on weekends. Due to discrepancies in the electronic record system of the two departments, only patients in the sedation center were enrolled in the current study. One anesthesiologist and six trained sedation nurses are assigned daily for sedation. Although the use of midazolam, sevoflurane, propofol, and dexmedetomidine has increased in recent years, most patients received chloral hydrate for non-invasive procedure sedation, due to the shortage of anesthesiologists. In the current study, only patients who received chloral hydrate as an initial sedative were included; no other pre-sedation hypnotics were given.

Chloral hydrate administration for sedation is relatively standard, with 25–100 mg/kg chloral hydrate (at a maximum dose of 2 g), depending on anesthesiologists’ evaluation. If the procedure cannot be completed with the initial dose, rescue medications are permitted, and the type and route of drug administration are determined by the anesthesiologist. In our institution, chloral hydrate redosing, midazolam, dexmedetomidine, or propofol can be selected as the rescue sedatives. If the rescue medication fails to complete the procedure, the anesthesiologist will decide whether to continue the third dose or postpone the examination. Patients are allowed to leave the sedation center when achieving a modified Aldrete score ≧9, which represents an awake patient.

The eligible patients were divided into two cohorts, morning or afternoon cases, according to the time of day when the initial dose of chloral hydrate was administered. The morning cases refer to the initial sedation being given from 7:00 A.M. to 11:59 A.M., and the afternoon cases were those in which the initial sedation was initiated between 12:00 P.M. and 17:00 P.M.

### 2.2. Data Extraction

Data were extracted from the electronic medical records, including gender, age, weight, source, type of procedures, sleep deprivation, the initial dose of choral hydrate, the sedation success of the initial dose, the sedation success rate with the second or the third dose, complications, and the length of recovery time.

### 2.3. Outcomes

The primary outcome was to assess whether the sedation success rate with an initial dose of chloral hydrate differed depending on the time of sedation (morning vs. afternoon). Sedation success was defined as patients being able to complete the procedures without a rescue dose.

The secondary outcomes are listed as follows:(1)The sedation success rate after the second dose;(2)Final sedation failure rate (defined as the patient not being able to finish the procedure, either because of the inability of achieving a sufficient depth of sedation or waking during procedures);(3)The duration of sedation was defined from chloral hydrate administration to awakening; only patients who achieved an anticipated depth of sedation after the initial dose and completed the procedure were included in the analysis.(4)Complications (vomiting, agitation, bradycardia, delayed awakening, rash, cough, hyperthermia, hypoxia, desaturation, and mild upper airway obstruction).(5)Subgroup analysis was conducted according to males and females.

### 2.4. Statistical Analysis

Because of the fasting requirements, younger children are usually scheduled in the morning and older children in the afternoon. Age differences, as one of the most important factors for circadian rhythms, may have had some impact on the outcomes. Therefore, propensity score matching was estimated using multivariate logistic regression, in which time of day of the sedation (morning vs. afternoon) was the dependent variable, and the matching ratio was 1:1, based on the nearest algorithms. The independent variables used in the multivariate logistic regression model were age, weight, type of patient (inpatients/outpatients), sleep deprivation, and procedures.

Categorical variables are presented as numbers and percentages. Continuous variables are reported as means (standard deviation (SD)) if normally distributed, otherwise reported as medians (interquartile range (IQR)). Differences in continuous variables were assessed using a Student *t*-test or Mann–Whitney U-test, as appropriate. Categorical variables were analyzed using Fisher’s exact test or a Chi-squared test. All statistical analyses were conducted using R studio (https://www.rstudio.com accessed on 28 November 2022; version 4.2.0). Statistical significance was considered as *p* < 0.05.

## 3. Results

### 3.1. Patient Characteristics

During the study period, 47,552 pediatric patients received chloral hydrate as an initiative sedation drug. Among them, 83 cases in which the time of chloral hydrate administration was unknown were not analyzed. Moreover, 5638 patients were excluded because of missing data. Finally, 41,831 patients were enrolled in the analysis. The study flow chart is shown in Figure 1. To avoid selection bias, the excluded cases (*n* = 5638) and analyzed cases (*n* = 41,831) were also compared. We found that the demographic characteristics and time of day distribution were similar between the two cohorts, except for the gender distribution (Appendix A).

Incomplete data were removed and the remaining 41,831 records were evaluated. Based on the published literature, the medication efficacy varied greatly between the morning and afternoon [13]. Therefore, the sedation success rate in pediatric patients before 12:00 PM and after 12 PM was compared (26,779 (64.0%) in the morning and 15,052 (36.0%) in the afternoon). Among them, 16,750 (40.0%) females and 25,081 (60%) males accepted sedation for non-painful procedures.

### 3.2. Before Propensity Score-Matching

Compared to the afternoon cases, the morning cases were younger (12.2 (4.0, 27.3) vs. 18.2 (6.0, 33.3), months, *p* < 0.01). Demographic characteristics and the distribution of procedures performed are shown in Table 1 and Figure 2. The median start time for morning cases was 9:08 A.M., with the earliest start time at 7:25 A.M. and the latest start time at 11:59 P.M., respectively; while in the afternoon, the median start time for afternoon cases was 13:23 P.M., and the earliest start time was at 12:00 P.M. and the latest start time was at 17:00 P.M. Overall, the sedation start time was trimodal, peaking at 8:00–9:00 A.M., 9:00–9:59 A.M., and 13:00–14:00 P.M. (Figure 3). Obviously, the patient demographic characteristics were not comparable between the morning and afternoon cases.

### 3.3. Primary and Secondary Outcomes before Matching

No statistical differences were detected in the sedation success rate with the initial dose of chloral hydrate between the morning cases and afternoon cases (morning cases: 23,238 (86.8%) vs. afternoon cases: 13,069 (86.8%), *p* = 0.90). The sedation success rate with a rescue dose of chloral hydrate was higher in the morning cases than those in the afternoon (morning cases: 1636 (6.1%) vs. afternoon cases: 711 (4.7%), *p* < 0.01). A similar trend was observed for the final sedation success rate (morning cases: 24,874 (92.9%) vs. afternoon cases: 13,780 (91.5%), *p* < 0.01). The recovery time was significantly longer in the morning cases than those sedated in the afternoon (morning cases: 65.0 (50.0, 84.0) vs. 60.0 (47.0, 76.0), minutes, *p* < 0.01).

### 3.4. Cohorts Matched by Propensity Score

As the children who received sedation in the morning were much younger than those who received sedation in the afternoon, it is predictable that the number of patients who could be matched was relatively small. Of the 15,052 patients undergoing sedation in the afternoon, 3907 patients were successfully matched to patients who accepted sedation in the morning. As a result, 7814 patients with similar pre-sedation demographic characteristics were available for analysis (Table 1).

### 3.5. Sedation Success Rate after Matching

In the matched cohorts, there was a statistically significant difference in the success rate of sedation with the initial dose of chloral hydrate between the morning and afternoon cases (morning cases: 86.1% (3364/3907) vs. afternoon cases 87.9% (3435/3907); *p* = 0.02). However, the mean initial dose of chloral hydrate was lower in the afternoon cases than those in the morning cases (Table 2).

The success rate of sedation with rescue medications was higher in the morning cases than those in the afternoon cases (morning cases: 6.2% (238/3907) vs. afternoon cases 4.4% (160/3907); *p* < 0.01) (Table 2). Rescue sedatives included an additional dose of chloral hydrate, midazolam, dexmedetomidine, and propofol. No significant differences in the distribution of rescue sedatives were detected between the two cohorts. Final sedation rates were comparable between the two groups (morning cases: 92.2% (3601/3907) vs. afternoon cases 92.0% (3223/3907); *p* = 0.62).

### 3.6. Sedation Duration and Complications after Matching

For sedation duration, the morning cases had a significantly longer median sedation duration compared to the afternoon cases (65.0 (51.0, 81.0) vs. 60.0 (48.0, 75.0), minutes, *p* < 0.01) (Table 2). Overall, the complications differed significantly, with patients receiving sedation services in the morning more likely to vomit than those who received sedation services in the afternoon.

### 3.7. Subgroup Analysis Based on Gender

Moreover, given that the published literature supports that circadian rhythms and mental functioning exhibit gender differences [15], we conducted a subgroup analysis to explore whether the sedation success rate differed in male and female children. As shown in Table 3, 2455 male pairs and 1402 female pairs were matched.

In males, the sedation success rate with the initial dose of chloral hydrate was significantly higher in the afternoon than those in the morning (2153 (87.7%) vs. 2094 (85.3%), *p* = 0.02); while in females, no difference was observed between morning and afternoon cases (1184 (84.5%) vs. 1221 (87.1%), *p* = 0.05). However, the sedation success rate with rescue medications was significantly higher in the morning cases than those in the afternoon cases for both genders (Table 3). No significant differences were detected in the final sedation failure or sedation duration between the morning and afternoon cases, either in the male or female groups.

## 4. Discussion

In this study, we evaluated the presence of daytime variation in the chloral hydrate sedation success rate with an initial dose. Due to unbalance demographic characteristics between morning and afternoon cases, a propensity score-matched method was used to minimize potential confounders. The results yielded three main findings. First, the success rate with the initial dose of chloral hydrate was higher in the patients sedated in the afternoon, despite the fact that the afternoon cases received a lower initial dose than the morning cases. Second, subgroup analysis showed that the success rate of sedation was higher in the afternoon for males than in the morning, but there was no difference between the morning and afternoon cases for females. Third, after subgroup analysis, the sedation duration was comparable among patients undergoing sedation services in the morning versus the afternoon, regardless of being male or female. No difference in the final sedation success rate was observed.

As aforementioned, chloral hydrate is still used in some countries, especially in nurse-led sedation services, and authors have emphasized that the use of chloral hydrate in structured pediatric nurse-led sedation protocols is safe and efficient [10]. However, the high failure rate of initial sedation with chloral hydrate is a source of parental dissatisfaction and should be taken seriously, as it imposes extra effort on medical staff and parents. According to the available literature, the sedation failure rate of chloral hydrate with an initial dose varied from 1.3 to 37% [10,16]. The sedation failure rate is inversely proportional to drug dose; for example, a fixed dose of 75 mg/kg chloral hydrate had a sedation failure rate of 1.3% [10], while that for 50 mg/kg was 24% [17]. However, adverse events associated with chloral hydrate were proportional to the dose of the drug, with adverse event rates of 11.1% at 50 mg/kg and 25% at 100 mg/kg [18]. Therefore, it is worthwhile exploring ways to improve the success rate of chloral hydrate sedation at relatively low doses. Although 25–100 mg/kg of chloral hydrate was permitted, 84% of patients were given doses equal to or lower than 50 mg/kg, in order to avoid adverse events.

Previous literature works reported that the patient’s age might have a potential influence on recovery time after sedation [13]. Thus, a propensity score-matched method, based on age and weight, was used to balance the demographic baseline. It is noted that younger infants were often scheduled at the earlier time and older children in the afternoon, which explains that despite 41,831 patients being identified during our study period, only 3907 pairs were successfully matched through the propensity-matched method.

Chronotherapeutics has been proposed in recent years, which involves synchronizing drug administration with a patient’s daily, monthly, seasonal, or yearly biological clock, in order to maximize health benefits and minimize adverse effects [19]. The time of day when a drug is administered can significantly change its efficacy. For example, several studies about hypertension management reported that the clinical therapeutic effect can be greatly improved when an anti-hypertension medication is administrated at bedtime rather than during morning hours [20]. However, less is known about chloral hydrate induced sedation, even though propofol and sevoflurane anesthesia have been well studied. Circadian rhythms are not established in neonates; thus, neonates were excluded from our study. We found that sedation success rates were higher when chloral hydrate was administered in the afternoon than in the morning. This may have been due to the consistency of timing of the patient’s own naps. Administration of sedatives coinciding with the onset of the patients’ normal sleep time may improve sedation, which was well summarized by Potts et al. [12]. In China, most children have the habit of taking afternoon naps, whether at home or in kindergarten. Afternoon naps have been proven to be good for learning and memory improvement [21,22].

As gender differences may be considered one of the potential factors influencing the results [13], a subgroup analysis was conducted, to explore if the chloral hydrate administered influenced the sedation success rate differently in male and female patients. Interestingly, the sedation success rate with the initial dose of chloral hydrate was higher in male subjects who accepted drugs during the afternoon than during the morning, whereas no differences were detected in females. However, this is hard to interpret based on our current knowledge, and further studies are needed to elucidate the mechanism.

To analyze the duration of chloral hydrate sedation, we enrolled patients who were successfully sedated. The results showed that the sedation duration with the initial dose of chloral hydrate among all patients in the morning was longer than those in the afternoon. However, the sedation duration was influenced by the degree of busyness of the sedation center. For example, children did not “waken” but were woken by medical staff to accelerate the turnaround. In addition, the business hours in the sedation center are from 8:00 A.M. to 17:00 P.M. In the afternoon, to finish work on time, patients may be woken artificially rather than naturally; while in the morning, most patients awaken naturally. This is also supported by our results, with longer sedations in the morning than in the afternoon. Surprisingly, after the subgroup analysis, the duration of sedation was slightly longer in morning cases compared to the afternoon cases, including males and females, but a difference that nearly reached statistical significance was detected.

## 5. Limitations

There are several obvious limitations. First, this is a retrospective study of over 40,000 cases. However, there were 5638 patients who could not be enrolled because of incomplete records. To reduce the selection bias, we compared the demographic characteristics between the included and excluded cases. No differences were detected in age, type of patient, weight, procedures, and distribution of chloral hydrate administration time, and only gender distribution had a significant difference between the two groups. Second, the nature of the retrospective study did not allow for adequate measurement of several potential confounding factors. For example, nurse personality might have some impact on sedation success. A patient nurse may be better at coaxing a child to take medicine, rather than forcing a child to take it while the child is crying. Patients who voluntarily take the medication may be more likely to fall asleep peacefully. Moreover, some patients woke up spontaneously after the procedure and did not return to the sedation center. Thus, we cannot document their awakening times, which explains why the number of children with a recorded sedation duration was less than the number of children with sedation success with the initial dose. Last, the reviewers and editors pointed out that the observed differences are not very marked, even though they are statistically significant. However, we believe that improvement of sedation efficiency is crucial for a busy institution. In our institution, about 100 patients receive sedation services per day; if one minute can be saved for each patient, more than one hour could be saved per day, which is conducive to reducing labor costs and improving parent satisfaction.

## 6. Conclusions

The results demonstrated a higher sedation success rate in the afternoon cases compared to the morning cases, despite the afternoon cases receiving a relatively lower initial dose. However, the clinical significance remains to be discussed, and further prospective studies are needed to validate the findings.

## Figures and Tables

**Figure 1 jcm-12-01245-f001:**
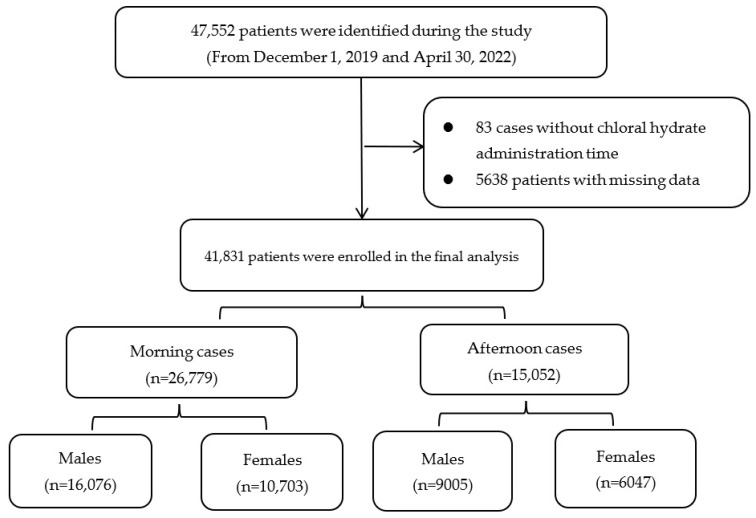
Study flow chart.

**Figure 2 jcm-12-01245-f002:**
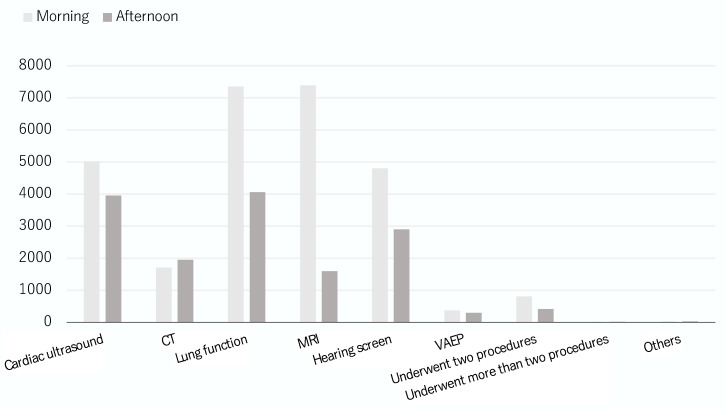
The distribution of procedures between the two cohorts.

**Figure 3 jcm-12-01245-f003:**
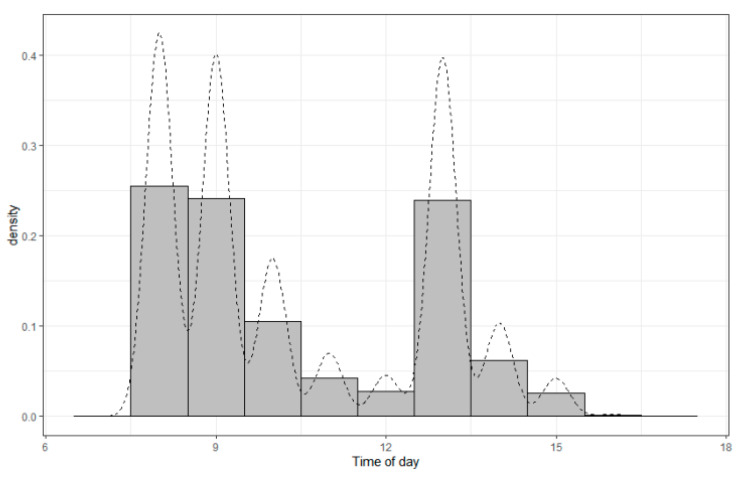
The sedation start time was trimodal, peaking at 8:00–9:00 A.M., 9:00–9:59 A.M., and 13:00–14:00 P.M.

**Table 1 jcm-12-01245-t001:** The demographic characteristics of the two cohorts.

	All Patients (*n* = 41,831)			Propensity Score Matched Pairs (*n* = 7816)		
	Morning Cases (*n* = 26,779)	Afternoon Cases (*n* = 15,052)	*p* Values	Morning Cases (*n* = 3907)	Afternoon Cases (*n* = 3907)	*p* Values
Male, *n* (%)	16,076 (60.0)	9005 (59.8)	0.69	2246 (57.5)	2327 (59.6)	0.07
Age, months, median (IQR)	12.2 (4.0, 27.3)	18.2 (6.0, 33.3)	<0.01 *	18.2 (6.0, 29.3)	18.2 (6.0, 29.3)	1.00
Weight, kg, median (IQR)	9.5 (6.5, 12.5)	10.5 (7.5, 13.5)	<0.01 *	10.5 (8.0, 13.0)	10.5 (8.0, 13.0)	1.00
Types of patient, outpatients, *n* (%)	16,982 (63.4)	9803 (65.1)	<0.01 *	2942 (75.3)	2942 (75.3)	1.00
Patients with sedation history, *n* (%)	9039 (33.8)	4577 (30.4)	<0.01 *	1082 (27.7)	1082 (27.7)	1.00
Sleep deprivation, yes, *n* (%)	11,629 (43.4)	6532 (43.4)	0.96	1921 (49.2)	1921 (49.2)	1.00
Procedures, *n* (%)			<0.01 *			1.00
Cardiac ultrasound	5013 (18.8)	3942 (26.2)		1099 (28.1)	1099 (28.1)	
CT	1687 (6.3)	1932 (12.8)		262 (6.7)	262 (6.7)	
Lung function	7345 (27.4)	4056 (26.9)		1219 (31.2)	1219 (31.2)	
MRI	6805 (25.4)	1533 (10.2)		395 (10.1)	395 (10.1)	
Hearing screen	4796 (17.9)	2895 (19.2)		917 (23.5)	917 (23.5)	
VAEP	343 (1.3)	260 (1.7)		6 (0.2)	6 (0.2)	
Others	26 (0.1)	27 (0.2)		0 (0.0)	0 (0.0)	
Underwent two procedures	750 (2.8)	396 (2.6)		9 (0.2)	9 (0.2)	
Underwent more than two procedures	14 (0.1)	11 (0.1)		0 (0.0)	0 (0.0)	

Note: CT (computed tomography), MRI (magnetic resonance imaging), VAEP (visual and auditory evoked potential); * *p* < 0.05.

**Table 2 jcm-12-01245-t002:** Primary and secondary outcomes.

	All Patients (*n* = 41,831)			Propensity Score Matched Patients (*n* = 7814)		
	Morning Cases (*n* = 26,779)	Afternoon Cases (*n* = 15,052)	*p* Values	Morning Cases (*n* = 3907)	Afternoon Cases (*n* = 3907)	*p* Values
**Primary outcome**						
Sedation success with initial dose, *n* (%)	23,238 (86.8)	13,069 (86.8)	0.90	3364 (86.1)	3435 (87.9)	0.02 *
Initial dose of chloral hydrate, mg/kg, median (IQR); Mean ± SD	50.0 (50.0, 50.0) 50.1 ± 2.2	50.0 (50.0, 50.0) 50.0 ± 2.0	0.25	50.0 (50.0, 50.0) 50.0 ± 1.5	50.0 (50.0, 50.0) 49.9 ± 1.4	0.01 *
**Secondary outcomes**						
Sedation success rate with a rescue dose, *n* (%)	1636 (6.1)	711 (4.7)	<0.01 *	238 (6.2)	160 (4.4)	<0.01 *
Rescue sedatives, *n* (%)			<0.01 *			0.53
Chloral hydrate	1200 (4.5)	478 (3.2)		171 (4.4)	110 (2.8)	
Midazolam	75 (0.3)	42 (0.3)		14 (0.4)	10 (0.3)	
Dexmedetomidine	354 (1.3)	185 (1.2)		57 (1.5)	38 (1.0)	
Propofol	7 (0.0)	6 (0.0)		1 (0.0)	2 (0.1)	
Final sedation success rate, *n* (%)	24,874 (92.9)	13,780 (91.5)	<0.01 *	3601 (92.2)	3594 (92.0)	0.80
Sedation duration, min, median (IQR)	63.0 (50.0, 80.5) *n* = 20,961	59.0 (47.0, 75.0) *n* = 11,591	<0.01 *	65.0 (51.0, 81.0) *n* = 2950	60.0 (48.0, 75.0) *n* = 3069	<0.01
Complications, *n* (%)						
Vomiting	406 (1.5)	154 (1.0)	<0.01 *	60 (1.6)	33 (0.9)	<0.01 *
Agitation	8 (0.0)	6 (0.0)	0.79	1 (0.0)	1 (0.0)	1.00
Cough	1 (0.0)	1 (0.0)	1.00	0 (0.0)	0 (0.0)	NA
Bradycardia	1 (0.0)	1 (0.0)	1.00	0 (0.0)	0 (0.0)	NA
Delayed awakening	2 (0.0)	1 (0.0)	1.00	0 (0.0)	0 (0.0)	NA
Hyperthermia	0 (0.0)	2 (0.0)	0.25	0 (0.0)	1 (0.0)	1.00
Rash	2 (0.0)	0 (0,0)	0.75	1 (0.0)	0 (0.0)	1.00
Desaturation	15 (0.1)	5 (0.0)	0.43	1 (0.0)	1 (0.0)	1.00

Note: NA, not applicable; * *p* < 0.01.

**Table 3 jcm-12-01245-t003:** Subgroup analysis based on gender.

**Male**						
	**Before Match (*n* = 25,081)**			**Propensity Score Matched Patients (*n* = 4910)**		
	**Before 12 P.M. (*n* = 16,076)**	**After 12 P.M. (*n* = 9005)**	***p*** **Values**	**Before 12 P.M.** **(*n* = 2455)**	**After 12 P.M. (*n* = 2455)**	***p*** **Values**
Age, months, median (IQR)	12.2 (4.4, 27.3)	19.1 (6.1, 34.3)	<0.01 *	19.2 (8.4, 30.3)	19.2 (8.4, 30.3)	1.00
Types of patient, outpatients, *n* (%)	10,302 (64.0)	5932 (65.9)	<0.01 *	1846 (75.2)	1846 (75.2)	1.00
Sleep deprivation, yes, *n* (%)	7036 (46.8)	3941 (47.4)	1.00	1211 (51.2)	1211 (51.2)	1.00
Procedures, *n* (%)			<0.01 *			1.00
Cardiac ultrasound	2865 (17.8)	2222 (24.7)		628 (25.6)	628 (25.6)	
CT	978 (6.1)	1152 (12.8)		158 (6.4)	158 (6.4)	
Lung function	4582 (28.5)	2556 (28.4)		873 (35.6)	873 (35.6)	
MRI	4147 (27.8)	933 (10.4)		257 (10.5)	257 (10.5)	
Hearing screen	2753 (17.1)	1725 (19.2)		525 (21.4)	525 (21.4)	
VAEP	216 (1.3)	157 (1.7)		6 (0.2)	6 (0.2)	
Others	15 (0.1)	11 (0.1)		0 (0.0)	0 (0.0)	
Underwent two procedures	511 (3.2)	240 (2.7)		8 (0.3)	7 (0.3)	
Underwent more than two procedures	9 (0.1)	9 (0.1)		0 (0.0)	0 (0.0)	
Success with initial dose, *n* (%)	14,022 (87.2)	7820 (86.8)	0.40	2094 (85.3)	2153 (87.7)	0.02 *
The initial dose of chloral hydrate, mg/kg, median (IQR)	50.0 (50.0, 50.0)	50.0 (50.0, 50.0)	0.28	50.0 (50.0, 50.0)	50.0 (50.0, 50.0)	0.09
Success with a rescue dose, *n* (%)	947 (5.9)	435 (4.8)	<0.01 *	163 (6.6)	111 (4.5)	<0.01 *
Rescue sedatives, *n* (%)			<0.01 *			0.51
Chloral hydrate	726 (4.5)	303 (3.4)		128 (5.2)	81 (3.3)	
Midazolam	45 (0.3)	27 (0.3)		3 (0.1)	4 (0.2)	
Dexmedetomidine	221 (1.4)	127 (1.4)		31 (1.3)	26 (1.1)	
Propofol	5 (0.0)	2 (0.0)		1 (0.0)	0 (0.0)	
Final sedation failure, *n* (%)	1107 (6.9)	750 (8.3)	<0.01 *	198 (8.1)	191 (7.8)	0.75
Sedation duration, min, Median (IQR)	65.0 (50.0, 84.0) (*n* = 12,635)	60 (47.0, 76.0) (*n* = 6960)	<0.01 *	64.0 (50.0, 80.0) (*n* = 1835)	60.0 (47.0, 75.0) (*n* = 1934)	0.07
**Female**						
	**Before Match (*n* = 16,750)**			**Propensity Score Matched Patients (*n* = 2804)**		
	**Before 12 P.M.** **(*n* = 10,703)**	**After 12 P.M. (*n* = 6047)**	***p*** **Values**	**Before 12 P.M.** **(*n* = 1402)**	**After 12 P.M. (*n* = 1402)**	***p*** **Values**
Age, months, median (IQR)	11.0 (4.0, 25.3)	17.2 (6.0, 32.3)	<0.01 *	17.2 (6.0, 28.3)	17.2 (6.0, 28.3)	1.00
Types of patient, outpatients, *n* (%)	6680 (62.4)	3871 (64.0)	0.04 *	1037 (74.0)	1037 (74.0)	1.00
Sleep deprivation, yes, *n* (%)	4590 (45.9)	2591 (46.6)	0.98	693 (52.0)	694 (49.5)	1.00
Sedation history, yes, *n* (%)	3547 (33.1)	1709 (28.3)	<0.01 *	512 (36.5)	425 (30.3)	0.26
Procedures, *n* (%)			<0.01 *			1.00
Cardiac ultrasound	2148 (20.1)	1720 (28.4)		466 (33.2)	466 (33.2)	
CT	709 (6.6)	780 (12.9)		96 (6.8)	96 (6.8)	
Lung function	2763 (25.8)	1500 (24.8)		377 (26.9)	377 (26.9)	
MRI	2658 (24.8)	600 (9.9)		124 (8.8)	124 (8.8)	
Hearing screen	2043 (19.1)	1170 (19.3)		335 (23.9)	335 (23.9)	
VAEP	127 (1.2)	103 (1.7)		2 (0.1)	2 (0.1)	
Others	11 (0.1)	16 (0.3)		0 (0.0)	0 (0.0)	
Underwent two procedures	239 (2.2)	156 (2.6)		2 (0.1)	2 (0.1)	
Underwent more than two procedures	5 (0.0)	2 (0.0)		0 (0.0)	0 (0.0)	
Success with initial dose, *n* (%)	9216 (86.1)	5249 (86.8)	0.22	1184 (84.5)	1221 (87.1)	0.05
The initial dose of chloral hydrate, mg/kg, Median (IQR)	50.0 (50.0, 50.0)	50.0 (50.0, 50.0)	0.61	50.0 (50.0, 50.0)	50.0 (50.0, 50.0)	0.08
Success with a rescue dose, *n* (%)	689 (6.4)	276 (4.6)	<0.01 *	76 (5.4)	51 (3.6)	0.03 *
Rescue sedatives, *n* (%)			0.12			0.74
Chloral hydrate	519 (4.8)	195 (3.2)		57 (4.1)	38 (2.7)	
Midazolam	31 (0.3)	15 (0.2)		3 (0.2)	1 (0.1)	
Dexmedetomidine	137 (1.3)	62 (1.0)		16 (1.1)	11 (0.8)	
Propofol	2 (0.0)	4 (0.1)		0 (0.0)	1 (0.1)	
Final sedation failure, *n* (%)	798 (7.3)	522 (8.6)	<0.01 *	142 (10.1)	130 (9.3)	0.48
Sedation duration, min, median (IQR)	65.0 (50.0, 82.0) (*n* = 8326)	60.0 (47.0, 75.0) (*n* = 4361)	<0.01 *	65.0 (51.0, 81.0) (*n* = 1032)	62.0 (50.0,77.0) (*n* = 1078)	0.65

Note: * *p* < 0.01.

## Data Availability

The datasets generated during and/or analyzed during the current study are available from the corresponding author on reasonable request.

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
