# Peer review of "Daytime Variation of Chloral Hydrate-Associated Sedation Outcomes: A Propensity-Matched Cohort Study"

_jcm, 2023, doi:10.3390/jcm12031245_

Round 1

Reviewer 1 Report

This is an interesting study of a common procedure in children involving a considerable number of patients.

It has ,of course ,has pointed out by the aughtors some limitations as it it is a retrospective study

Line 50 --abreviation PACU is used without an explanation

As indicated minor editing is required with the english in relation to tenses

Examples  lines 51,93 and 138 use of 12 .00 pm .The time is either Noon (as in this caase) or Midnight.

Lines 230-231 the word received is used on two occasions which causes difficulty in comprehension.

Line 289 the children did not "waken but were wakened

Author Response

Reviewer 1:

This is an interesting study of a common procedure in children involving a considerable number of patients.

It has ,of course ,has pointed out by the aughtors some limitations as it it is a retrospective study

Line 50 --abbreviation PACU is used without an explanation

Answer: Thank you for your suggestion. An explanation for the abbreviation PACU has been added.

Szolnoki et al. retrospectively analyzed 2340 children who underwent brain MRI under general anesthesia and found a strong relationship between time of day and the length of post-anesthetic care unit PACUstay. (Page 2, line 51-54)

As indicated minor editing is required with the english in relation to tenses

Answer: English has polished by the native speaker. The certification is attached.

Examples  lines 51,93 and 138 use of 12 .00 pm .The time is either Noon (as in this case) or Midnight. 

Answer: Thank you. This is a good question. To avoid confusion, we define the morning cases and the afternoon cases in the methods section.

“The morning cases were referred to the initial sedation started from 7:00 AM to 11:59 AM, and the afternoon cases were those in which the initial sedation was initiated between 12:00 PM and 17:00 PM.The 12:00 PM is noon instead of midnight.  

Lines 230-231 the word received is used on two occasions which causes difficulty in comprehension.

Answer: This sentence is revised as follows.

First, the success rate with the initial dose of chloral hydrate was higher in the patients sedated in the afternoon, despite the fact that the afternoon cases received a lower initial dose than the morning cases. (Page 9, line 239-241)

Line 289 the children did not "waken but were wakened

Answer: We have revised accordingly.

“For example, the children did not “waken” but were wakened by medical staff to accelerate the turnaround. ” (Page 10, line 297-298)

Reviewer 2 Report

Review of Manuscript ID jcm-2097409: “Daytime Variation Of Chloral Hydrate Associated Sedation Outcomes: A Propensity-Matched Cohort Study” submitted to Journal of Clinical Medicine, section Pharmacology by Yu Cui et al.

That was an interesting manuscript (retrospective study, including 41,831 cases) which is dedicated to assessing the circadian rhythms on chloral hydrate sedation success rate in pediatric patients (except neonates). Additionally, subgroup analyses were provided according to gender. Some secondary outcomes were presented as well, including, the chloral hydrate safety, the sedation success rates with a second dose; the duration of sedation with chloral hydrate; and final sedation failure rates. Overall the paper was well written, studies and analysis were well organized. My specific comments are listed below.  

Minor Comments:

1.      I would like to make just one suggestion. I think it will be better if the authors include a piece of information about the mechanism of action of chloral hydrate and its pharmacokinetics. This helps for a better understanding of its effects and side effects.

2.      In line 154 one bracket is missing after p<0.01.

3.      In line 316 There is “6” at the end of the conclusion. Probably it’s a mistake.

Author Response

Reviewer 2:

That was an interesting manuscript (retrospective study, including 41,831 cases) which is dedicated to assessing the circadian rhythms on chloral hydrate sedation success rate in pediatric patients (except neonates). Additionally, subgroup analyses were provided according to gender. Some secondary outcomes were presented as well, including, the chloral hydrate safety, the sedation success rates with a second dose; the duration of sedation with chloral hydrate; and final sedation failure rates. Overall the paper was well written, studies and analysis were well organized. My specific comments are listed below.  

Answer: Thank you for your valuable comments. 

Minor Comments:

  1. I would like to make just one suggestion. I think it will be better if the authors include a piece of information about the mechanism of action of chloral hydrate and its pharmacokinetics. This helps for a better understanding of its effects and side effects.

Answer: Thank you for your valuable suggestions. In the background section, we have include a piece of information about the mechanism of action of chloral hydrate and its pharmacokinetics as follows.

Chloral hydrate is a nonopiate, nonbenzodiazepine sedative-hypnotic drug, which is absorbed from the gastrointestinal tract, and rapidly converted to the pharmacologically active metabolite, trichloroethanol (TCE), which is responsible for the sedative properties [1].

  1. Pershad J, Palmisano P, Nichols M. Chloral hydrate: the good and the bad. Pediatr Emerg Care. 1999;15(6):432-435.

  1. In line 154 one bracket is missing after p<0.01.

Answer: Thank you. The missing bracket has been added.

  1. In line 316 There is “6” at the end of the conclusion. Probably it’s a mistake.

Answer: Thank you. Yes, it is a mistake. The number “6” has been removed.
